# Copper-catalyzed dehydrogenative $\gamma$-C(sp$^3$)-H amination of saturated ketones for synthesis of polysubstituted anilines

Rong Hu[1,2], Fa-Jie Chen[2], Xiaofeng Zhang[2], Min Zhang[2] & Weiping Su [1,2]

Metal-catalyzed $\beta$-C-H functionalization of saturated carbonyls via dehydrogenative desaturation proved to be a powerful tool for simplifying synthesis of valuable $\beta$-substituted carbonyls. Here, we report a copper-catalyzed dehydrogenative $\gamma$-C(sp$^3$)-H amination of saturated ketones that initiates the three-component coupling of saturated ketones, amines and $N$-substituted maleimides to construct polysubstituted anilines. The protocol presented herein enables both linear and $\alpha$-branched butanones to couple a wide spectrum of amines and various $N$-substituted maleimides to produce diverse tetra- or penta-substituted anilines in fair-to-excellent yields with good functional group tolerance. The mechanism studies support that this ketone dehydrogenative $\gamma$-C(sp$^3$)-H amination was triggered by the ketone $\alpha,\beta$-dehydrogenation desaturation that activates the adjacent $\gamma$-C(sp$^3$)-H bond towards functionalization. This $\alpha,\beta$-dehydrogenation desaturation-triggered cascade sequence opens up a new avenue to the remote C(sp$^3$)-H functionalization of saturated ketones and has the potential to enable the rapid syntheses of complex compounds from simple starting materials.

[1] School of Physical Science and Technology, ShanghaiTech University, Shanghai 201210, China. [2] State Key Laboratory of Structural Chemistry, Center for Excellence in Molecular Synthesis, Fujian Institute of Research on the Structure of Matter, Chinese Academy of Sciences, Fuzhou 350002, China. Correspondence and requests for materials should be addressed to W.S. (email: wpsu@fjirsm.ac.cn)

Ketones, a large class of versatile and readily available substrates, conventionally participate in reactions with their electrophilic *ipso-* carbons or nucleophilic $\alpha$-carbons[1]. Recently, the $\beta$-functionalization reactions of saturated ketones have been achieved by employing directing group-assisted Pd-catalyzed $C(sp^3)$–H activation methods[2–6], or the strategies of merging photoredox catalysis with organocatalysis[7,8], or metal-catalyzed ketone $\alpha,\beta$-dehydrogenation desaturation/the resultant enone coupling cascade sequence[9–18]. Despite these advances in the development of the approaches to reactions at $\beta$-carbons of ketones, for the direct $\gamma$-$C(sp^3)$–H functionalization of saturated ketone, only a few of examples have been reported to date[2,19–21]. The first example is the Pd-catalyzed $\gamma$-arylation reaction of the ketone lacking any $\beta$-$C(sp^3)$–H bond using glycine as a transient directing group (Fig. 1a), which was limited to only a single ketone substrate[2]. The other two examples for ketone $\gamma$-$C(sp^3)$–H functionalization both involved the use of $\alpha$-imino-oxy acids prepared from the condensation of ketones with $\alpha$-aminoxy acids as starting materials: one was the auxiliary-assisted Pd-catalyzed $\gamma$-$C(sp^3)$–H arylation reaction of $\alpha$-imino-oxy acid with aryl iodides that afforded $\gamma$-arylated ketones after Mn-catalyzed removal of $\alpha$-imino-oxy acid auxiliary (Fig. 1b)[19]; the other was the photoredox-catalyzed cross-coupling between $\alpha$-imino-oxy acid and radical trapping reagents that furnished, after treatment of the reaction products with water, the $\gamma$-functionalized ketones as final products (Fig. 1c)[20,21]. Notably, the metal-catalyzed remote $C(sp^3)$–H functionalization, which is similar to ketone $\gamma$-$C(sp^3)$–H functionalizaion in terms of the distance between the reaction position and functional group, has been accessible to a handful of carboxylic acid derivatives containing specific auxiliary directing groups[22–27]. These reactions of carboxylic acid derivatives disclosed that their appropriate auxiliary directing groups were essential for remote $C(sp^3)$–H functionalization. However,

the relatively low reactivity of ketones severely limits the scope of the directing groups that are pre-installed or in situ installed to saturated ketone frameworks, and therefore poses a great challenge to the development of transient directing group strategy for metal-catalyzed $\gamma$-$C(sp^3)$–H functionalization of simple ketones. The metal-catalyzed methods for direct $\gamma$-functionalization of saturated ketones without the need for pre-installation of auxiliary directing group to ketone framework is highly desired, given that such methods would expand the reactivity patterns of ketones, streamline syntheses of value-added $\gamma$-functionalized ketones.

Metal-catalyzed dehydrogenative $\beta$-$C(sp^3)$–H functionalization reactions via dehydrogenative desaturation[28,29] were also amenable to esters[30–34], lactams[35], and other substrates[36–48], demonstrating the ability of this type of reaction to rapidly construct complex molecule from simple substrates. Such reactions will continue to flourish in the future in view of the recent advance in the development of catalytic methods for generation of unsaturated compounds via dehydrogenation[39–49] as well as the versatility of unsaturated compounds as synthetic intermediates. Recently, we have discovered a Cu-catalyzed successive dehydrogenation of aliphatic ketones to furnish dienketones or polyenketones[50]. This successive dehydrogenation sequence started with $\alpha,\beta$-dehydrogenation desaturation to form enone intermediates, and therefore implicated that the carbonyl group activated $\gamma$-$C(sp^3)$–H bond of enone through $\alpha,\beta$-carbon–carbon double bond, as occurred in organocatalyzed $\gamma$-functionalization of enals[51–54] and $\beta$-functionalization of alkyl aldehydes[55–57] in which organocatalyst moieties covalently linked to substrates through formyl group exerted an effect on the remote C–H bonds through carbon–carbon double bonds. Inspired by our previous findings, we envisioned the feasibility of the $\alpha,\beta$-desaturation initiated $\gamma$-functionalization of aliphatic ketones with nucleophilic

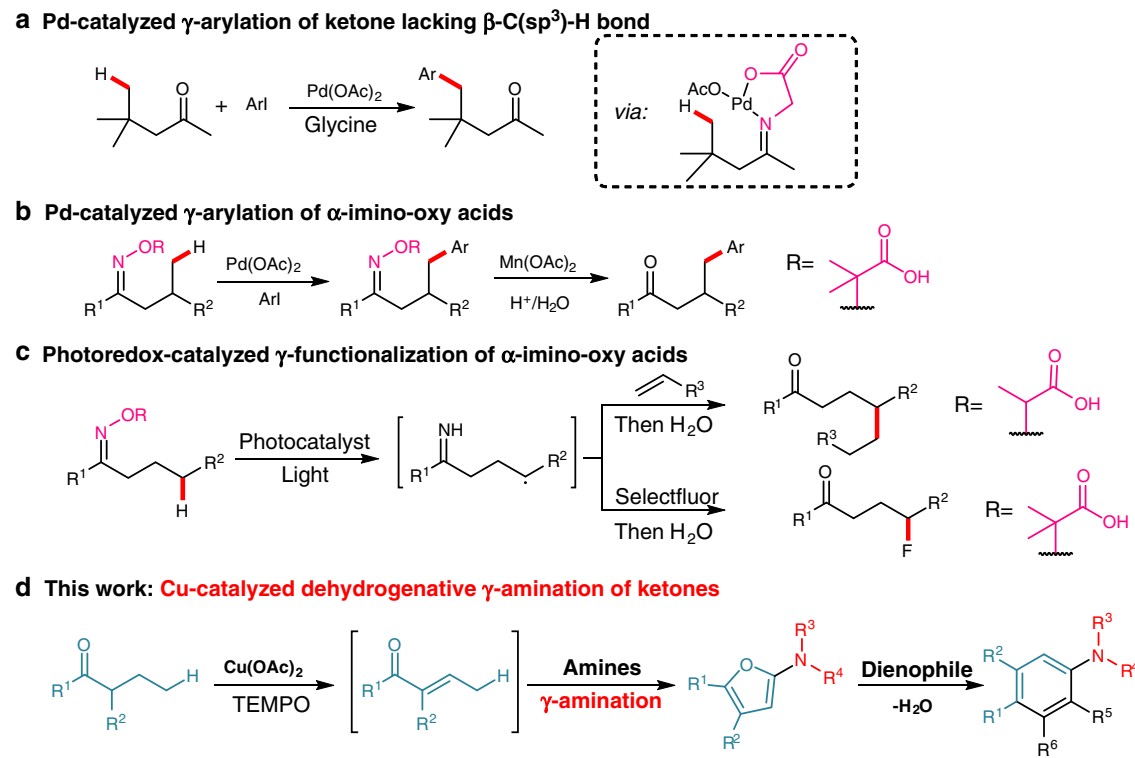

**a** Pd-catalyzed $\gamma$-arylation of ketone lacking $\beta$-C(sp³)-H bond

**b** Pd-catalyzed $\gamma$-arylation of $\alpha$-imino-oxy acids

**c** Photoredox-catalyzed $\gamma$-functionalization of $\alpha$-imino-oxy acids

**d** This work: Cu-catalyzed dehydrogenative $\gamma$-amination of ketones

**Fig. 1** $\gamma$-C(sp³)–H functionalization of ketones. **a** Pd-catalyzed $\gamma$-arylation reaction of the ketone using glycine as a transient directing group. **b** Auxiliary-assisted Pd-catalyzed $\gamma$-C(sp³)–H arylation reaction of $\alpha$-imino-oxy acid. **c** Photoredox-catalyzed $\gamma$-C(sp³)–H functionalization of $\alpha$-imino-oxy acid with radical trapping reagents. **d** Our work proposed that $\alpha,\beta$-desaturation initiated $\gamma$-functionalization of ketones

amines that was expected to take place through $\alpha,\beta$-desaturation of ketone to enone intermediate, subsequent oxidation of $\gamma$-C ($sp^3$)–H bond of enone to form electrophilic species, final capture of the resulting electrophilic species with amine to form C–N bond at the position $\gamma$ to carbonyl group. This targeted dehydrogenative ketone $\gamma$-amination is challenging to access because the electron-withdrawing carbonyl disfavors the oxidation of enone $\gamma$-C($sp^3$)–H bond to generate electrophilic species[58], mechanistically contrasting with the previously reported secondary amine-catalyzed $\gamma$-functionalization of enal that is, in principle, the dehydrogenative cross-coupling of enal with electrophilic reagent via deprotonation of $\gamma$-C($sp^3$)–H in the iminum intermediate of enal to form nucleophilic species[51,52,54,59,60].

Herein, we report a Cu-catalyzed dehydrogenative $\gamma$-amination reaction of saturated ketone that initiates a 2-amino furan formation/[4 + 2] cycloaddition of resultant 2-amino furan with N-substituted maleimide cascade sequence toward construction of polysubstituted anilines (Fig. 1d). The mechanism studies reveals that this Cu-catalyzed $\gamma$-amination reaction is triggered by ketone $\alpha,\beta$-dehydrogenative desaturation, which activates the $\gamma$-C($sp^3$)–H bond of the resultant enone intermediate toward amination. Our findings demonstrate that the $\alpha,\beta$-dehydrogenative desaturation triggered ketone $\gamma$-functionalization is viable even using nucleophilic amines as coupling partners. Since these polysubstituted anilines are readily converted to diverse valuable compounds[61,62], this operationally simple, efficient method for the syntheses of polysubstituted anilines from simple substrates will attract the attentions from chemists working in a variety of research fields.

## Results

**Discovery and development of reaction.** To check our hypothesis, we initially investigated the reaction of 1-phenyl-1-butanone (**1a**) (2 equiv.) with diisopropylamine (**2a**) in toluene at 120 °C with 10 mol% Cu(OAc)$_2$/10 mol% 2,2′-bipyridine (bpy) as a catalyst and 2,2,6,6-tetramethylpiperidine-N-oxyl (TEMPO)[63,64] (3 equiv.) as an

oxidant, and gratifyingly found that the expected ketone $\gamma$-amination reaction did occur, but afforded 2-diisopropylamino-5-phenylfuran (**6a**) as a final product. Due to partial decomposition of electron-rich 2-amino furan during the work-up stage, 2-amino-5-phenylfuran was isolated in only moderate yield (about 45%). Thus, we attempted to use N-substituted maleimide as a dienophile to capture the reactive 2-amino furan generated from ketone $\gamma$-amination and obtain polysubstituted aniline as a final product[65] since polysubstituted anilines are highly useful compounds and generally accessed via multistep synthetic sequences[66]. To our delight, once introducing N-ethyl maleimide as the third component to the reaction system, three-component cross-coupling occurred to furnish tetra-substituted aniline in a 80% isolated yield (Table 1, entry 1), providing a step- and atom-economic approach to synthesis of polysubstituted aniline from simple starting materials. Optimization studies for this three-component cross-coupling reaction (Table 1) revealed that acids as an additive affected the reaction outcome. Totally, 0.2 equiv. of ortho-nitro-benzoic acid slightly increased the yield to 85% (Table 1, entry 2), while weaker carboxylic acids led to decrease in yield (Table 1, entries 5 and 6). Although acetic acid had a negative effect on the reaction, a solution of p-TsOH in acetic acid benefited the reaction. The beneficial effect of strong acid p-TsOH relied on its amount (Table 1, entry 8 versus entries 7 and 10). Totally, 0.1 equiv. of p-TsOH gave the best result (Table 1, entry 8). p-TsOH monohydrate was inferior to a solution of p-TsOH in acetic acid (Table 1, entry 9 versus entry 8), implicating the detriment of water to the reaction. As revealed by the control experiments, the reaction gave the desired product in the diminished yields in the absence of Cu catalyst (Table 1, entries 11–13), while the reaction was almost shut down on removing both Cu catalyst and TsOH (Table 1, entry 14). The results of control experiments indicated that both Cu catalyst and TsOH facilitated the targeted reaction. A solvent was also observed to affect the reaction outcome (Table 1, entry 2 versus entries 3 and 4).

---

**Table 1 Selected results of reaction optimization[a]**

| Entry | Additive (equiv.) | Solvent | Yield(%)[b] |
|---|---|---|---|
| 1 | – | PhCH$_3$ | 80 |
| 2 | o-nitro-benzoic acid (0.2) | PhCH$_3$ | 85 |
| 3 | o-nitro-benzoic acid (0.2) | o-DCB | 67 |
| 4 | o−nitro-benzoic acid (0.2) | CH$_3$CN | 63 |
| 5 | benzoic acid (0.2) | PhCH$_3$ | 53 |
| 6 | acetic acid (0.2) | PhCH$_3$ | 56 |
| 7 | p-TsOH (0.05) | PhCH$_3$ | 87 |
| 8 | p-TsOH (0.1) | PhCH$_3$ | 96 |
| 9 | p-TsOH•H$_2$O (0.1) | PhCH$_3$ | 83 |
| 10 | p-TsOH (0.2) | PhCH$_3$ | 91 |
| 11[c] | p-TsOH•H$_2$O (0.08) | PhCH$_3$ | 30 |
| 12[c] | p-TsOH (0.1) | PhCH$_3$ | 33 |
| 13[c] | p-TsOH (0.2) | PhCH$_3$ | 21 |
| 14[c] | – | PhCH$_3$ | trace |

[a]Reaction conditions: **1a** (2 equiv.), **2a** (0.4 mmol), **3a** (1.5 equiv.), Cu(OAc)$_2$ (10 mol%), bpy (10 mol%), TEMPO (3 equiv.), additive, solvent (1.5 mL), N$_2$, 120 °C for 48 h
[b]Isolated yields
[c]Without Cu(OAc)$_2$/bpy

---

**Fig. 2** Scope of ketones[a]. [a]Reaction conditions: **1** (2 equiv.), **2a** (0.4 mmol), **3a** (1.5 equiv.), Cu(OAc)$_2$ (10 mol%), bpy (10 mol%), TEMPO (3 equiv.), p-TsOH (10 mol%), PhCH$_3$ (1.5 mL), N$_2$, 120 °C for 48 h. Isolated yields

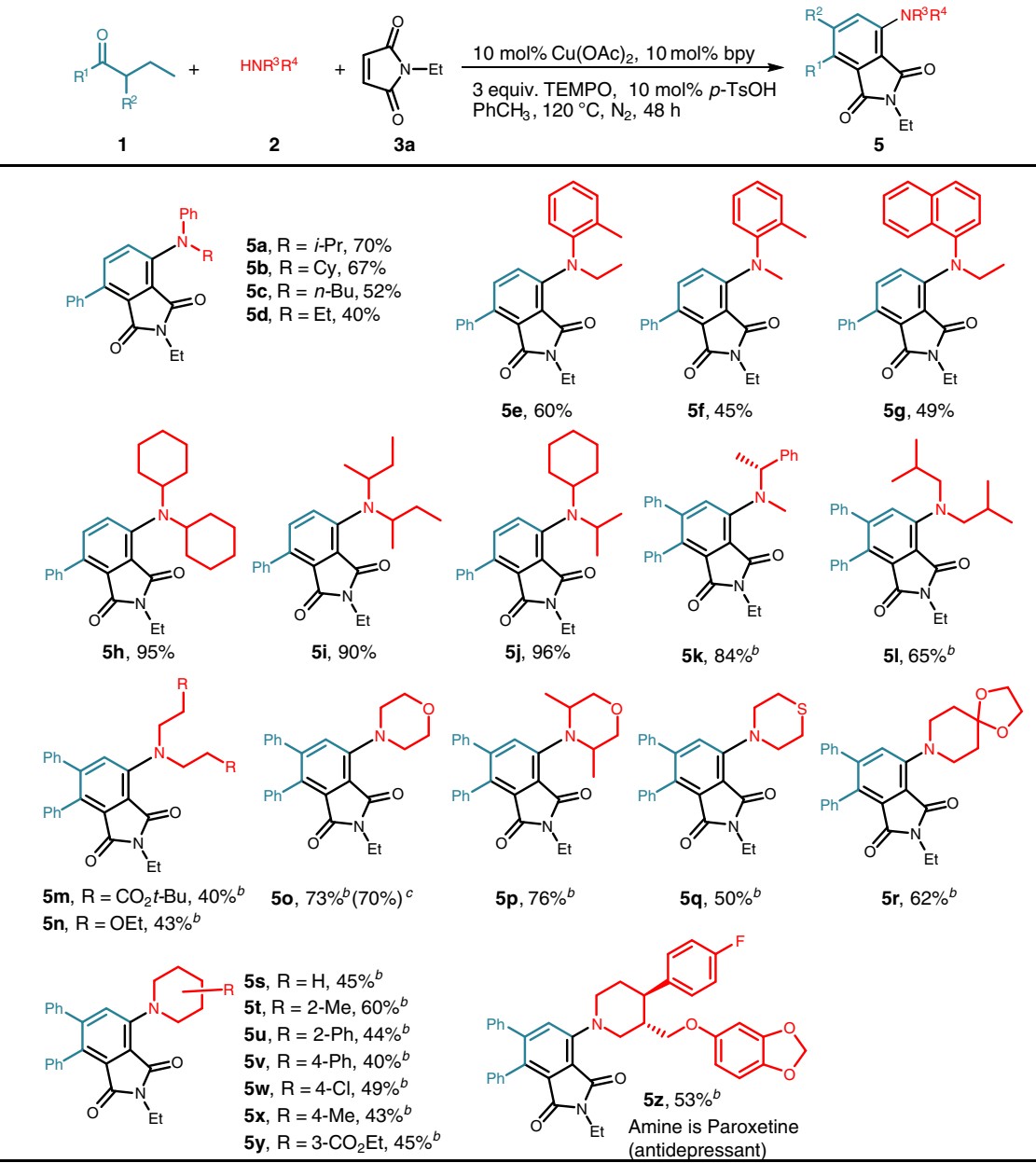

**Fig. 3** Scope of secondary amines[a]. [a]Reaction conditions: **1** (2 equiv.), **2** (0.4 mmol), **3a** (1.5 equiv.), Cu(OAc)$_2$ (10 mol%), bpy (10 mol%), TEMPO (3 equiv.), p-TsOH (10 mol%), PhCH$_3$ (1.5 mL), N$_2$, 120 °C for 48 h. Isolated yields. [b]Reaction conditions: **1** (0.4 mmol), **2** (0.4 mmol), Cu(OAc)$_2$ (10 mol%), bpy (10 mol%), TEMPO (3 equiv.), CsOAc (20 mol%), PhCH$_3$ (1.5 mL), N$_2$, 120 °C for 38 h; then **3a** (1.5 equiv.) is added for another 10 h. [c]Reaction was performed on 7.14 mmol scale

**Substrate scope of ketones**. With the optimized reaction conditions in hand, we explored the substrate scope with respect to ketones. As shown in Fig. 2, 1-aryl-1-butanones bearing diverse substituents on their phenyl rings smoothly underwent reactions to give the corresponding polysubstituted anilines in excellent yields (**4a–4g**) with the exception of cyano-substituted ketone (**4e**) that probaly coordinated to Cu catalyst through cyano group to intervene with Cu-catalysis. α-Substituted 1-aryl-1-butanones (**4h–m**), such as butanones containing β-diketone (**4j, 4k**), β-keto-ester (**4i, 4l, 4m**) moieties, participated in the reaction to furnish penta-substituted anilines in moderate-to-excellent yields. Among these substrates, α-acetyl-substuted 1-phenyl-1-butanone gave two regioisomers with near 1: 1.6 ratio (**4k**). 1-Heteroaryl-1-butanones proved to be suitable substrates for this reaction, as exemplified by 1-(furan-2-yl)butan-1-one (**4n**) and 1-(thiophen-

2-yl)butan-1-one (**4o**). Both α-substituent-lacking and α-sub-stituent-containing dialkyl ketones also underwent the poly-substituted aniline formation reaction occurring exclusively on the propyl moiety (**4p–w**). 1-Phenylhexan-3-one gave styrenyl-substitued target product as a result of over-dehydrogenation (**4s**). In spite of a lot of attempt, β-substituted ketone did not work for this reaction presumably due to the difficulty in α,β-dehydrogenation desaturation. The established reaction conditions were also amenable to the three-component coupling reaction of but-2-enal, as shown by 2-phenylbut-2-enal (**4x**), which provided a complement to the existing method for enal γ-functionalization[51–54].

**Substrate scope of secondary amines**. Figure 3 demonstrates that the Cu-catalyzed three-component coupling reaction was

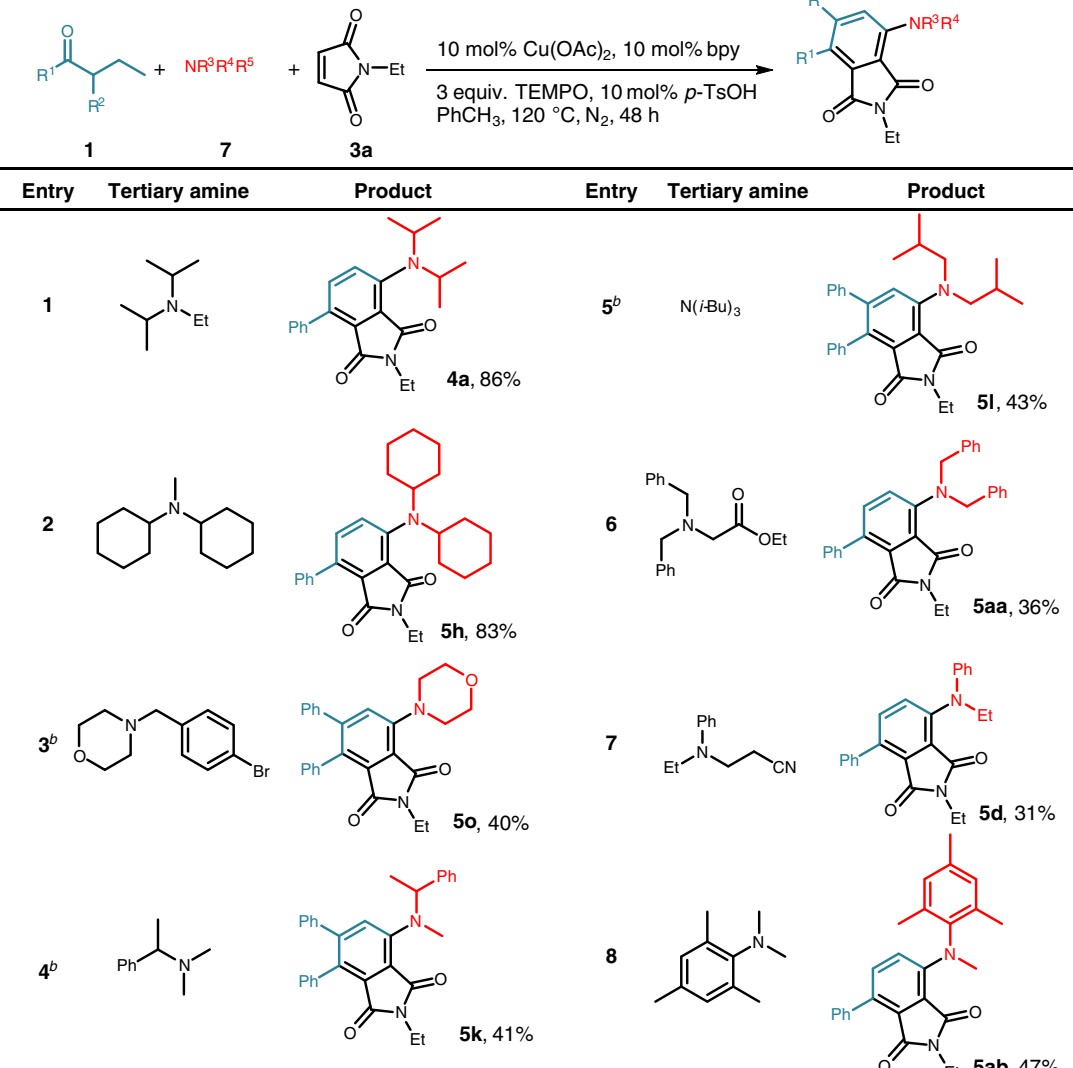

**Fig. 4** Scope of tertiary amines[a]. [a]Reaction conditions: **1** (0.4 mmol), **7** (2 equiv.), **3a** (1.5 equiv.), Cu(OAc)$_2$ (10 mol%), bpy (10 mol%), TEMPO (3 equiv.), p-TsOH (10 mol%), PhCH$_3$ (1.5 mL), N$_2$, 120 °C for 48 h. Isolated yields. [b]Reaction conditions: **1** (0.4 mmol), **2** (2 equiv.), Cu(OAc)$_2$ (10 mol%), bpy (10 mol%), TEMPO (3 equiv.), CsOAc (20 mol%), PhCH$_3$ (1.5 mL), N$_2$, 120 °C for 38 h; Then **3a** (1.5 equiv.) is added for another 10 h

compatible with a broad range of secondary amines using 1-phenyl-1- butanone or α-phenyl-substituted 1-phenyl-1- butanone as a coupling partner. N-alkyl anilines furnished the corresponding products in good yields (**5a–g**). Besides, acyclic dialkyl amines afforded products in fair-to-excellent yields (**5h–5n**). Notably, an array of alicyclic amines with varying substitutents and substitution patterns, including methyl (**5t** and **5x**), phenyl (**5u** and **5v**), chloro (**5w**), ester (**5y**) groups were well tolerated in this reaction. Thiomorpholine (**5q**) was also suitable substrate and provided synthetically useful yields. To demonstrate the practicability of our method, the gram-scale reaction of **1h** (1.61 g, 7.14 mmol) was conducted to afford the corresponding product (**5o**) in 70% yield (2.06 g). The three-component coupling reaction was also capable of modifying paroxetine containing the alicyclic amine moiety (**5z**), illustrating its potential in practical utilities.

**Substrate scope of tertiary amines**. Interestingly, tertiary amines could participate in this three-component coupling reaction via cleavage of C–N bond[67,68], thus play the similar role to secondary amines (Fig. 4). Although the exact reason for the cleavage of

C–N bond remains to be clarified, among the investigated tertiary amines, the C–N bond cleavage occurred preferentially either at more sterically accessible carbon atoms (entries 1, 2, 4, and 8) or at carbon atoms adjacent to substituents capable of stabilizing carbon radicals (entries 3 and 6). For example, diisopropyl ethylamine gave excellent yield (entry 1) via cleavage of C–N bond of less sterically hindered ethyl group while bulkier triiso-butyl amines gave moderate yield (entry 5).

In addition, this three-component coupling reaction was compatible with the variation of substituent on nitrogen atom of maleimide. A series of N-substituents of maleimide, such as benzyl, aryl and alkyl groups, were tolerable for the reaction (See Supplementary Methods for details).

**Mechanistic studies**. To gain an insight into this Cu-catalyzed dehydrogenative γ-amination of saturated ketone, we performed mechanistic investigations. The reaction of 1-phenyl-1-butanone (**1a**) with TEMPO under standard conditions was observed to produce 2,2,6,6-tetramethyl-1-(5-phenylfuran-2-yl) piperidine (**6b**) as a result of in situ generation of 2,2,6,6-tetramethyl piperidine from TEMPO (eq. 1, Fig. 5), which prevented us from

**a  Identification of reaction intermediates**

**b  Investigation of the role of Cu species**

**c  Proposed reaction pathway**

**Fig. 5** Preliminary studies of mechanism. **a** The control experiments show that the cascade sequence may proceed through initial ketone $\alpha,\beta$-dehydrogenation desaturation that activates the adjacent $\gamma$-C(sp³)–H bond. **b** The control experiments show that Cu catalyst is the main contributor to the ketone $\alpha,\beta$-dehydrogenation desaturation step and p-TsOH catalyze the conversions of enone intermediates. **c** The proposed reaction pathway to generate polysubstituted anilines

identifing the intermediates from oxidation of saturated ketone by TEMPO. Despite this, on reducing reaction temperature to 110 °C, 1-phenylbut-2-en-1-one (**6c**) and 4-oxo-4-phenylbut-2-enal (**6d**) were identified along with formation of **6b** in the reaction of **1a** with TEMPO (eq. 2, Fig. 5). Both enone **6c** and γ-keto enal **6d** were observed to react with diisopropyl amine under standard conditions to produce 2-diisopropylamino-5-phenyl-furan (**6a**) (eqs. 3 and 4, Fig. 5). Identification of **6c** and **6d** in the reaction of **1a** with TEMPO, in combination with their conversion to 2-diisopropylamino furan (**6a**), suggested that both **6c** and **6d** could be intermediates in the reaction of saturated 1-phenyl-1-butanone with diisopropyl amine to generate 2-amino-furan. Moreover, the reaction of enone **6c** with TEMPO under milder conditions (110 °C) was observed to generate γ-keto enal **6d** and 2-amino furan **6b** (eq. 5, Fig. 5). Generation of **6d** from the reaction of **6c** with TEMPO illustrated that enone **6c** was likely the precursor to γ-keto enal **6d**. In light of our previous findings that γ-TEMPO substituted enone was found in the reaction of enone with TEMPO[50] and that α-C(sp$^3$)–H of imine was oxidized by TEMPO to C=O bond via α-TEMPO substituted imine intermediate[69], conversion of enone **6c** to γ-keto enal **6d** likely stemmed from the γ-oxygenation of enone **6c** with TEMPO, in which 2,2,6,6-tetramethyl piperidine released from TEMPO was incorporated into **6b**.

In the optimization studies, control experiments showed that p-TsOH could facilitate the reaction, but was less effective than Cu catalyst alone. In the reaction of 1-phenyl-1-butanone (**1a**) with diisopropylamine to form **6a** (eq. 6, Fig. 5), the simultaneous use of Cu catalyst and p-TsOH gave 45% yield while 16% yield was obtained in the absence of Cu catalyst, which is consistent with the results obtained in optimization studies. In contrast, in the reactions of enone **6c** with diisopropylamine (eq. 3, Fig. 5) or with diisopropylamine and N-ethyl maleimide (eq. 7, Fig. 5), p-TsOH alone gave the comparable yields to those obtained with the simultaneous use of Cu catalyst and p-TsOH. These results implicated that Cu catalyst was more active than p-TsOH in the catalysis of ketone α,β-dehydrogenation desaturation step and that p-TsOH as a catalyst was effective in the conversions of enone intermediates.

On the basis of the above investigations, a cascade sequence was proposed for the three-component coupling reaction (Fig. 5c). The cascade sequence may proceed through initial α,β-dehydrogenation desaturation to form α,β-enone, γ-C(sp$^3$)-H oxidation of resultant α,β-enone to a γ-keto enal intermediate, amine–aldehyde condensation between γ-keto enal with secondary amine to form 2-amino furan via intramolecular cyclization of iminium species, and final cycloaddition of 2-amino furan to N-substituted maleimide to afford polysubstituted aniline after dehydration aromatization.

## Discussion

In summary, a Cu-catalyzed three-component coupling of saturated ketones, amines and N-substituted maleimides via dehydrogenative γ-C(sp$^3$)–H amination of saturated ketones has been developed for syntheses of polysubstituted anilines. The dehydrogenative γ-C(sp$^3$)–H amination of saturated ketones was triggered by the ketone α,β-dehydrogenation desaturation that activates the adjacent γ-C(sp$^3$)–H bond. This α,β-dehydrogenation desaturation triggered sequence opens up a new avenue to the remote C(sp$^3$)–H functionalization of saturated ketones, and has the potential to enable the rapid syntheses of complex compounds from readily available saturated ketones in a single operation. Efforts to develop other dehydrogenative γ-C(sp$^3$)–H functionalization reactions[28,29,70] of ketones through clarifying reaction mechanism are underway in our group.

## Methods

**General procedure for ketones with secondary amines reaction**. In a nitrogen-filled glovebox, a 25 mL Schlenk tube equipped with a stir bar was charged with Cu (OAc)$_2$ (7.26 mg, 0.04 mmol, 10 mol%), 2,2′-bipyridine (6.25 mg, 0.04 mmol, 10 mol%), N-substituted maleimide (0.6 mmol, 1.5 equiv.) and TEMPO (187.50 mg, 1.2 mmol, 3.0 equiv.). The tube was fitted with a rubber septum and moved out of the glove box. Then amine (0.4 mmol), ketone (0.8 mmol), p-Toluenesulfonic acid (6.88 mg, 0.04 mmol, 10 mol%, 12 wt% solution in pure acetic acid), and toluene (1.5 mL) were added in turn to the Schlenk tube through the rubber septum using syringes, and the septum was replaced with a Teflon screwcap under nitrogen flow. The reaction mixture was allowed to stir for 48 h at 120 °C. After completion of the reaction, the reaction mixture was cooled to room temperature. Then the reaction mixture was diluted with ethyl acetate (10 mL), followed by filtration through a pad of silica gel with several washings. Then the filtrate was concentrated under reduced pressure, and purified by flash column chromatography on silica gel to provide the corresponding product.

**General procedure for ketones with tertiary amines reaction**. In a nitrogen-filled glovebox, a 25 mL Schlenk tube equipped with a stir bar was charged with Cu (OAc)$_2$ (7.26 mg, 0.04 mmol, 10 mol%), 2,2′-bipyridine (6.25 mg, 0.04 mmol, 10 mol%), TEMPO (187.50 mg, 1.2 mmol, 3 equiv.), and N-substitutedmaleimide (75.08 mg, 0.6 mmol, 1.5 equiv.). The tube was fitted with a rubber septum and moved out of the glove box. Then ketone (0.4 mmol), tertiary amine (0.8 mmol, 2.0 equiv.), p-Toluenesulfonic acid (6.88 mg, 0.04 mmol, 10 mol%, 12 wt% solution in pure acetic acid) and toluene (1.5 mL) were added in turn to the Schlenk tube through the rubber septum using syringes, and then the septum was replaced with a Teflon screwcap under nitrogen flow. The reaction mixture was stirred at 120 °C for 48 h. Upon cooling to room temperature, the reaction mixture was diluted with 10 mL of ethyl acetate, followed by filtration through a pad of silica gel with several washings. The filtrate was concentrated under reduced pressure, and then purified by flash column chromatography on silica gel to provide the corresponding product.

## Data availability

All data generated and analyzed during this study are included in this article and its Supplementary Information files, and are also available from the authors on reasonable request. Crystallographic data have been deposited at the Cambridge Crystallographic Data Centre (CCDC), under deposition number 1905935 (**4a**), 1905936 (**6b**), and can be obtained free of charge from the CCDC via http://www.ccdc.cam.ac.uk/data_request/cif.

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

## Acknowledgements

This work was supported by the National Key Research and Development Program of China (2017YFA0206801), National Natural Science Foundation of China Grants (21431008, u1505242, and 21702205), the Strategic Priority Research Program of the Chinese Academy of Sciences (XDB20000000 and XDB10040304) and the Key Research Program of the Chinese Academy of Sciences (ZDRW-CN-2016-1).

## Author contributions

R.H. and W.S. conceived the project. R.H. performed the experiments and analyzed the data. F.-J.C. and M.Z. discussed the results. X.Z. characterized X-ray structures of two compounds. W.S. prepared the paper.

## Additional information

**Competing interests:** The authors declare no competing interests.

