## [Peer Review File · Nature Communications]

Reviewers' comments:

Reviewer #1 (Remarks to the Author):

Su and coworkers developed Cu/TEMPO-Catalyzed Dehydrogenation-Conjugate Addition for β -Functionalization of Saturated Ketones in 2016 (ref. 14) and then the similar Cu/TEMPO system was applied to the consecutive dehydrogenation or dehydrogenation of various saturated ketones, aldehydes, alcohols, α,β -unsaturated diesters, and N-heterocycles in 2017 (ref. 52).

In this manuscript, by using the similar Cu/TEMPO system, they realized Dehydrogenative γ -Amination of Saturated Ketones followed by a 2-amino furan formation and [4+2] cycloaddition cascade sequence to construct anilines. Even though the catalytic system is not novel anymore, the γ -amination with both secondary and tertiary amines is interesting. The three-component coupling realized construction of diverse multi-substituted anilines. Thus, the referee recommends publish after minor revisions. Some questions for consideration that would be of particular interest, include:

- (1) Please give some explanation for the role of p-TsOH. Could the authors speculate as to which step is benefited? Why were acids screened in this optimization study? Will other bases, like DMAP or DBU, have an effect in this reaction?
- (2) The order of substrate scopes in Fig 2 is confusing. Grouping them based on R1 or R2 may make the figure to be more easily read. Such as 4u should be grouped with 4i, 4l, 4m.
- (3) In Fig 4, why was CsOAc added for entries 3, 4, and 5? What happened when 3a was added at the beginning of the reaction? Under what circumstances should these alternative procedures be employed?
- (4) In Fig 5, 6b was generated in 50% yield without addition of amines. Were 6b or TMP-aniline formed as byproducts for other substrates under standard conditions?
- (5) If the N-substituted maleimide was changed to other kinds of dienophiles, would this reaction still work? The authors should comment on this scope even if not successful. What about other maleimides (N-benzyl or alkyl) or maleic anhydride?
- (6) Were beta-substituted substrates attempted? This should be discussed, as it would lead to fully substituted aniline products.

Reviewer #2 (Remarks to the Author):

This manuscript describes the development of a 3-component coupling reaction that produces highly substituted anilines as products.

The authors use their previously-described conditions for desaturation of ketones to form enones in situ. These enones are then activated toward reaction with amines and maleimide as catalyzed by acid and potentially copper. The idea of using desaturation to activate the gamma-position of ketones is novel and could influence the thinking of others in this field. That said, the authors preface their paper by discussing the difficulty with gamma-functionalization of ketones. The chemistry included in this paper is not a solution to that problem since the product is not a gamma-functionalized ketone. While the products are interesting and potentially useful, they are narrowly applicable. The mechanistic studies are also preliminary and do not provide enough insight to stimulate research in this area.

While the chemistry introduced is novel and potentially useful in very specific situations, the reactions are not broadly applicable. In such a case, I expect compelling mechanistic insight into unique reactivity for a paper to warrant publication in Nature Communications. However, the mechanistic studies only provide expected outcomes and do not provide definitive insight that might inspire this field. In my opinion this paper should be published in a high quality journal with a more specific audience. Before publishing the manuscript elsewhere, the authors should address the following.

The role of copper is unclear since TsOH has a significant catalytic effect. To clarify this the authors should report the results of reaction using anhydrous TsOH at 10 mol %. They report the results with hydrated TsOH, which is inferior as a catalyst. The authors should also report 20 mol% anhydrous TsOH.

To further clarify the role of Cu, the authors should perform the experiment in eq 1 using 6c, but without Cu/bpy.

The authors should provide evidence that 4k maintains the benzylic stereochemistry.

Reviewer #3 (Remarks to the Author):

this is an interesting reaction, although not exactly C-H activation, and this does not provide method to specifically functionalize a single C-H bond site selectively. however the cascade strategy traps the dehydrogenation product is a good one and the product is useful. the conditions are interesting finding.

I can support this for publication. the table and scheme are not clear as to what ketones are compatible, looks like mostly aromatic ketones are compatible.

Point-by-point responses to comments of reviewers

(Manuscript ID: NCOMMS-19-12829, by Hu et al.)

1. Responses to the comments of Reviewer #1

Comment #1.

Su and coworkers developed Cu/TEMPO-Catalyzed Dehydrogenation-Conjugate Addition for β -Functionalization of Saturated Ketones in 2016 (ref. 14) and then the similar Cu/TEMPO system was applied to the consecutive dehydrogenation or dehydrogenation of various saturated ketones, aldehydes, alcohols, α,β -unsaturated diesters, and N-heterocycles in 2017 (ref. 52).

In this manuscript, by using the similar Cu/TEMPO system, they realized Dehydrogenative γ -Amination of Saturated Ketones followed by a 2-amino furan formation and [4+2] cycloaddition cascade sequence to construct anilines. Even though the catalytic system is not novel anymore, the γ -amination with both secondary and tertiary amines is interesting. The three-component coupling realized construction of diverse multi-substituted anilines. Thus, the referee recommends publish after minor revisions. Some questions for consideration that would be of particular interest, include:

Response: We thank the referee for his/her positive comments on our work.

Comment #2: *(1) Please give some explanation for the role of p-TsOH. Could the authors speculate as to which step is benefited? Why were acids screened in this optimization study? Will other bases, like DMAP or DBU, have an effect in this reaction?*

Response: During the revision of this manuscript, we carried out mechanism studies of the Cu-catalysed three-component coupling reaction. Our mechanism investigations revealed that this reaction was a cascade reaction involving the initial α,β -dehydrogenation desaturation of ketone to enone, subsequent γ -C (sp^3)-H oxidation of enone to generate γ -keto enal, aldehyde-amine condensation induced cyclization to form 2-amino furan, final [4+2] cycloaddition of 2-amino furan to N-substituted maleimide to afford polysubstituted aniline after dehydration aromatization. Control experiments revealed that TsOH itself could promote the targeted reaction without copper species, albeit in much lower yield than that obtained with Cu(OAc)₂/bpy. In the experiments for mechanistic insights, TsOH was observed to facilitate the reaction of enone, diisopropylamine and N-ethyl maleimide to produce polysubstituted aniline in high yield in the absence of Cu(OAc)₂/bpy. On the basis of these results, TsOH may be effective for the γ -C (sp^3)-H oxidation of enone to γ -keto enal step, but reluctantly promote α,β -

dehydrogenation desaturation. In addition, TsOH may also facilitate aldehyde-amine condensation step and [4+2] cycloaddition step. In response to this comment, I discussed the role of TsOH in the part of mechanism study of the revised manuscript.

Considering that acid may promote [4+2] cycloaddition of 2-amino furan as well as the subsequent dehydration aromatization, we screened effects of acids and their amounts on the reaction outcomes in the early optimization studies.

A variety of bases have been screened to check their effects on the model reaction, and the results from these experiments have added to Table S1 in supplementary material. We observed that some of bases such as CsOAc slightly improved the yield in the model reaction while other bases such as DBU and DMAP had a negative effect on the reaction.

Comment #3: (2) The order of substrate scopes in Fig 2 is confusing. Grouping them based on R1 or R2 may make the figure to be more easily read. Such as 4u should be grouped with 4i, 4l, 4m.

Response: Thank the referee for this suggestion. In the revised figures 2, 3 and 4, the different structural moieties of products, which come from the corresponding reactants, have been marked with specific colors.

Comment #4: (3) In Fig 4, why was CsOAc added for entries 3, 4, and 5? What happened when 3a was added at the beginning of the reaction? Under what circumstances should these alternative procedures be employed?

Response: In the cases of entries 3, 4 and 5 in Fig. 4, we found that the optimized reaction conditions did not give the satisfied yield. To obtain useful yields, we further screened the reaction conditions for these substrates and found that CsOAc improved the reaction yields. The beneficial effect of CsOAc was also observed in the reaction of other less bulky amines (please see Fig 3.), consistent with the optimization studies for the model reaction in which CsOAc was observed to slightly improve the reaction yield. For less bulky amines, adding **3a** to the reaction system at the beginning of the reaction led to the undesired nucleophilic addition of amine to *N*-ethyl maleimide (**3a**), and reduced the yield of target product. To avoid this side reaction of less bulky amines, **3a** was added after the reaction of saturated ketones with amines was run for 38 hours.

Comment #5: (4) In Fig 5, 6b was generated in 50% yield without addition of amines. Were 6b or TMP-aniline formed as byproducts for other substrates under standard conditions?

Response: Under standard conditions, the targeted three-component coupling reactions of the external amines outcompete the formation of **6b** side-product likely because the concentration of 2,2,6,6-tetramethyl piperidine (TMP) *in-situ* generated from TEMPO is much lower than those of external amine substrates and the sterical hindrance of TMP retards its reaction. As a result, in the three-component reactions, none of **6b** side-product was observed.

Comment #6: (5) *If the N-substituted maleimide was changed to other kinds of dienophiles, would this reaction still work? The authors should comment on this scope even if not successful. What about other maleimides (N-benzyl or alkyl) or maleic anhydride?*

Response: Thank the referee for this question. The substrate scope with *N*-substituted maleimide has been investigated. Other maleimides containing a series of *N*-substituents such as benzyl, aryl and alkyl groups proved to be suitable coupling partners. Other kinds of dienophiles such as maleic anhydride did not serve as suitable coupling partners under standard conditions because of side-reaction of amine with these dienophiles. In response to this comment, we have described the scope and limitation of maleimides in the revised manuscript, and added five more examples with typical *N*-substituted maleimides to the supplementary material.

Comment #7: (6) *Were beta-substituted substrates attempted? This should be discussed, as it would lead to fully substituted aniline products.*

Response: we have checked two β -substituted ketones for this three-component coupling reaction and found that these β -substituted ketones did not work for this reaction likely because of the difficulty in α,β -dehydrogenation desaturation of these ketones. In the revised manuscript, we have discussed this issue.

2. Responses to the comments of Reviewer #2

Comment #1: *This manuscript describes the development of a 3-component coupling reaction that produces highly substituted anilines as products.*

The authors use their previously-described conditions for desaturation of ketones to form enones in situ. These enones are then activated toward reaction with amines and maleimide as catalyzed by acid and potentially copper. The idea of using desaturation to activate the gamma-position of ketones is novel and could influence the thinking of others in this field. That said, the authors preface their paper by discussing the difficulty with gamma-functionalization of ketones. The chemistry included in this paper is not a solution to that problem since the product is not a gamma-functionalized ketone. While the products are interesting and potentially useful, they are narrowly applicable. The

mechanistic studies are also preliminary and do not provide enough insight to stimulate research in this area.

Response: We appreciate the referee for his/her positive comments on our work, especially for the comment that “*The idea of using desaturation to activate the gamma-position of ketones is novel and could influence the thinking of others in this field*”. Indeed, inspired by our previous work on the consecutive dehydrogenation (please see: ref. 52), we explored the possibility that the α,β -dehydrogenation of saturated ketone may induce its γ -C(sp³)-H functionalization. The three-component coupling reaction described herein showcased the γ -C(sp³)-H amination of saturated ketones, and validated our proposal to some extent, though the reported reaction gave polysubstituted anilines rather than γ -substituted ketones.

We agree with the referee’s comment “*The mechanistic studies are also preliminary and do not provide enough insight to stimulate research in this area*”. Thus, we have done more experiments to investigate the reaction mechanism. The reaction of 1-phenyl-1-butanone (**1a**) with TEMPO under standard conditions was observed to produce 2,2,6,6-tetramethyl-1-(5-phenylfuran-2-yl) piperidine (**6b**) as a result of *in-situ* generation of 2,2,6,6-tetramethyl piperidine from TEMPO (eq. 1, Fig. 5), which prevented us from identifying the intermediates from oxidation of saturated ketone by TEMPO. Despite this, on reducing reaction temperature to 110 °C, 1-phenylbut-2-en-1-one(**6c**) and 4-oxo-4-phenylbut-2-enal (**6d**) were identified along with formation of **6b** in the reaction of **1a** with TEMPO (eq. 2, Fig. 5). Both enone **6c** and γ -keto enal **6d** were observed to react with diisopropyl amine under standard conditions to produce 2-diisopropylamino-5-phenyl-furan (**6a**) (eqs. 3 and 4, Fig. 5). Identification of **6c** and **6d** in the reaction of **1a** with TEMPO, in combination with their conversion to 2-diisopropyl amino furan **6a**, suggested that both **6c** and **6d** could be intermediates in the reaction of saturated 1-phenyl-1-butanone with diisopropyl amine to generate 2-amino-furan. Moreover, the reaction of enone **6c** with TEMPO under milder conditions (110 °C) was observed to generate γ -keto enal **6d** and 2-amino furan **6b** (eq 5, Fig. 5). Generation of **6d** from the reaction of **6c** with TEMPO illustrated that enone **6c** was likely the precursor to γ -keto enal **6d**. In light of our previous findings that γ -TEMPO substituted enone was found in the reaction of enone with TEMPO (please see: Ref. 52) and that α -C(sp³)-H of imine was oxidized by TEMPO to C=O bond via α -TEMPO substituted imine intermediate (Please see: Nat. Commun. 2018, 9, 5002), conversion of enone **6c** to γ -keto enal **6d** likely stemmed from the γ -oxygenation of enone **6c** with TEMPO, in which 2,2,6,6-tetramethyl piperidine released from TEMPO was incorporated into **6b**. These mechanistic investigations allowed us to rationalize the dehydrogenative coupling of ketone with amine, and provided evidences to support our proposal that α,β -dehydrogenation desaturation of saturated ketones could trigger their γ -C(sp³)-H functionalization. In view of the versatile reactivity of formyl group, diverse

dehydrogenative γ -functionalization reactions of saturated ketones via γ -keto enal intermediates can be designed.

Comment #2: *While the chemistry introduced is novel and potentially useful in very specific situations, the reactions are not broadly applicable. In such a case, I expect compelling mechanistic insight into unique reactivity for a paper to warrant publication in Nature Communications. However, the mechanistic studies only provide expected outcomes and do not provide definitive insight that might inspire this field. In my opinion this paper should be published in a high quality journal with a more specific audience. Before publishing the manuscript elsewhere, the authors should address the following.*

Response: We appreciate the referee for his/her suggestion on the deeper mechanism studies. To obtain insights into the mechanism, we identified the possible reaction intermediates and investigated the conversions of these intermediates. In addition, we explored the roles of copper species and TsOH in the reaction (see below). These mechanistic investigations have allowed us to rationalize the cascade sequence for 2-amino furan formation starting from saturated ketone and amines, and supported our proposal that α,β -dehydrogenation desaturation of saturated ketones could activate their γ -C(sp³)-H bonds towards oxidative functionalization.

Comment #3: *The role of copper is unclear since TsOH has a significant catalytic effect. To clarify this the authors should report the results of reaction using anhydrous TsOH at 10 mol %. They report the results with hydrated TsOH, which is inferior as a catalyst. The authors should also report 20 mol% anhydrous TsOH.*

Response: Thank the referee for this suggestion. To clarify the role copper salt in the targeted reaction, we carried the model reaction (the reaction in Table 1) with varying amounts of anhydrous TsOH in the absence of copper salt and bpy ligand. Without copper salt and bpy ligand, 10 mol% anhydrous TsOH gave 33% yield, and increasing amount of anhydrous TsOH to 20 mol% instead reduced the yield to 21%. Model reaction gave only a trace of product in the absence of both TsOH and copper salt. These control experiments showed that TsOH enabled accelerating the reaction, but was less effective than copper catalyst alone (please see entry 1 vs. entry 12, Table 1). These experimental results have been added to Table 1 in the revised manuscript.

Comment #4: *To further clarify the role of Cu, the authors should perform the experiment in eq 1 using 6c, but without Cu/bpy.*

Response: The reaction of enone **6c** with diisopropylamine (in the revised manuscript, the reaction in eq. 3) in the absence of Cu/bpy gave lower yield (35%) than that with Cu/bpy (48%), while the reaction of **6c**, diisopropylamine and *N*-ethyl maleimide in the absence of Cu/bpy gave polysubstituted aniline in 89% yield (eq 7, Fig. 5). In the reaction of saturated 1-phenyl-1-butanone **1a** with diisopropylamine, the 45% yield of 2-aminofuran obtained with 10 mol% Cu(OAc)₂/10 mol% bpy was much higher than that obtained without Cu/bpy (16%) (eq 6, Fig. 5). These experimental results indicated that copper species plays a key role in the promotion of α,β -dehydrogenation desaturation of ketones to enone while TsOH was efficient for acceleration of the reaction of enone. These results and the corresponding discussion have added the part of mechanistic studies in the revised manuscript.

Comment #5: *The authors should provide evidence that 4k maintains the benzylic stereochemistry.*

Response: We guess that the referee referred to **5k** rather than **4k** because only **5k** contains chiral center. To get its absolute configuration, we attempted to get its crystal for single crystal X-ray diffraction. Unfortunately, our attempt failed because its flexible carbon chain make it difficult to grow single crystals of good quality.

3. Responses to the comments of Reviewer #3

Comment #1: *this is an interesting reaction, although not exactly C-H activation, and this does not provide method to specifically functionalize a single C-H bond site selectively. however the cascade strategy traps the dehydrogenation product is a good one and the product is useful. the conditions are interesting finding. I can support this for publication. the table and scheme are not clear as to what ketones are compatible, looks like mostly aromatic ketones are compatible.*

Response: We appreciate the referee for his/her positive comment and good suggestion. To clarify what ketones are compatible, in the revised figures 2, 3 and 4, the different structural moieties of products, which come from the corresponding reactants, have been marked with specific colors.

REVIEWERS' COMMENTS:

Reviewer #1 (Remarks to the Author):

The authors have addressed all referee concerns, and is ready for publication.

Response to the comments of Reviewer # 1

Comment #1: The authors have addressed all referee concerns, and is ready for publication.

Response: we appreciate the reviewer for his/her positive comment. We are happy that our revisions could properly address his/her concerns.